# Surface Treatment of Mongolian Scots Pine Using Phosphate Precipitation for Better Performance of Compressive Strength and Fire Resistance

**DOI:** 10.3390/ma16072711

**Published:** 2023-03-29

**Authors:** Yan Ge, Liang Wang, Xuepeng Wang, Hao Wang

**Affiliations:** School of Civil Engineering and Architecture, Anhui University of Science and Technology, Huainan 232001, China

**Keywords:** wood, phosphate, pore structure, mechanical properties, heat release, smoke production, fire resistance

## Abstract

Wood, as a naturally green and environmentally friendly material, has been widely used in the construction and decoration industries. However, the flammability of wood poses serious safety problems. To improve the fire resistance of wood, In this study, it is proposed to use calcium chloride (CaCl_2_) and disodium hydrogen phosphate (Na_2_HPO_4_, DSP) to impregnate wood for multiple cycles. The experimental results show that phosphate mineral precipitation can be deposited on the surface of the wood. Scanning electron microscopy (SEM) and X-ray diffraction (XRD) are used to analyze the micromorphology of mineral precipitation and use the MIP test to analyze the treated wood pore structure. The results show that with the increase in the number of cycles, the phosphate deposited on the surface of the wood increases, and the cumulative pore volume and water absorption rate of the wood after 10 cycles are 54.3% and 13.75% lower than that of untreated wood respectively. In addition, the cone calorimeter (CONE) confirmed that the total heat release (THR) and total smoke production (TSP) of wood treated in 10 cycles have decreased by 48.7% and 54.2% respectively compared with the untreated wood. Hence, this treatment method not only improves the mechanical properties of wood. It also improves fire resistance.

## 1. Introduction

Wood is a renewable natural resource with high strength, low density, and good processing properties, which has been an important building material since ancient times. Compared to reinforcement, concrete, and other building materials, wood is more environmentally friendly with good thermal insulation properties [1,2]. Thus, wood can be widely processed into load-bearing members such as beams and columns, as well as nonload-bearing members such as partition walls and partitions [3]. The main components of wood are cellulose, hemicellulose, and lignin with elements of C, H, and O, which also determines that wood is a flammable biomass material [4,5,6,7]. In the event of a fire, the wooden components in the building are prone to burning, posing a great threat to the building and its users [8].

To overcome the flammability of wood, some retardant treatment methods had been developed, such as the impregnation method, surface modification method, sol-gel method, and spraying method [9,10,11]. Impregnation is the surface treatment method, including atmospheric impregnation and vacuum-pressurized impregnation [12,13]. Atmospheric pressure impregnation is to use the impregnated flame retardant to penetrate the interior of the wood spontaneously at normal pressure and temperature. Wang et al. [14]. selected phytic acid to treat wood at atmospheric pressure. Research shows that the total heat release of treated wood can be reduced by 71%. The coke content of the combusted sample has been increased from 12.9% to 34.3%, which can form the protection matrix with a higher accumulation density and inhibit further combustion inside the wood. Vacuum pressure impregnation is to first extract the air in the micropores of the wood, then use the air pressure difference to inhale the flame retardant. Li et al. [15] compared the effect of organic solution impregnation and inorganic solution impregnation on the performance of wood. They used phenol formaldehyde oligomer and water glass, respectively, to modify the wood by vacuum pressure impregnation. Compared to untreated wood, the total heat release of treated wood was significantly decreased, indicating that the fire resistance of wood has been improved. Moreover, the effect of inorganic impregnation is better than that of organic impregnation. Wang et al. [16] used acrylate and functional monomers containing phosphorus, nitrogen, and sulfur as raw materials to develop a new type of wood flame-retardant coating. The retardant coating can be coated on the wood surface by thiol-UV curing, which greatly improves the fire resistance of wood. Girardi et al. [17] impregnated pine blocks in sols prepared from vinyl acetic acid, isopropyl alcohol, and vinyl trimethoxy silane. The gels and polymers can be coated on the wood surface, which not only improves the fire resistance of the wood, it also helps the wood resist fungi such as brown rot. Li et al. [18] successfully synthesized a new phosphate epoxy reaction flame retardant using a one-step reaction of vanillin (VA) and benzene phosphorous oxydichloride (BPOD) and then coated the wood. Through the conical calorimeter test, the total heat release (THR) and the smoke production (TSP) treated were 47.95 MJ/m^2^ and 6.26 m^2^/kg respectively, which were 50.8% and 71.2% lower than untreated wood. With the continuous expansion of the application field of wood, research on the mechanical properties of wood has gradually increased. While pursuing the fire resistance of wood, it is also hoped that the mechanical properties can be improved. Sun et al. [19] used sodium silicate, aluminum sulfate, and organic epoxy resin to form a Si-Al-EP over a branched structure inside the wood under the crosslinking of triethanolamine. Compared with untreated wood, it not only improves the fire resistance of wood, it also increases the flexural strength and compressive strength. Zhang et al. [20] used the decomposition and expansion of sodium bicarbonate to destroy the internal structure of poplar. Then, a vacuum impregnated a high-concentration sodium silicate solution into pores, so that silicon dioxide can be formed inside and fill the internal pore structure of the wood. The results show that the fire resistance of poplar after treatment is improved compared with untreated poplar. The compressive strength and bending strength were increased by 86.32% and 96.27%, respectively. Zhang et al. [21] studied the impact of ammonium polyphosphate (APP) and silicon dioxide (SiO_2_) on the flammability and mechanical properties of wood-fiber–PP composites. The results show that both APP and silicon dioxide can improve the fire resistance of wood-fiber–PP composites and reduce the initial temperature of thermal degradation. Silicon dioxide has a flame-retardant synergy with APP in wood fiber–PP composites.

Current research shows that these treatment methods are beneficial to wood fire resistance. However, some treatment methods used toxic reagents, which not only harm operators, they also cause unpredictable harm to users. In addition, high-dose flame retardants may cause inefficiency and high cost. Therefore, the treatment method should be simple and easy to operate. The treated wood should be harmless to the environment and human beings. 

In response to the above problems, and the increasing demand for sustainable use of biomass products, as well as people’s concerns about environmental and health problems, the use of toxic flame retardants is increasingly avoided and restricted [22,23,24]. Some halogenated flame retardants are being phased out. The current research of relevant researchers on green, environmentally friendly flame retardants has gradually increased and the development field of flame retardants has gradually increased. The relevant scholars are committed to finding green, environmentally protecting fuel boosters. It can be seen that the development of more nontoxic and harmless degradable flame retardants has become the ultimate goal [25,26,27,28,29].

This study develops a novel surface treatment method to improve the fire resistance of wood by using calcium phosphate precipitations. Those precipitations can be deposited on the surface of the wood and fill the voids through a cyclic immersion in the calcium ion and phosphate solutions under normal temperature and pressure. The treatment process of this method is simple and easy to operate. It not only improves the fire resistance of wood, it also generated hydroxyapatite precipitation that can improve the acid resistance and corrosion resistance of wood. The precipitated components on the wood surface were analyzed by X-ray diffraction (XRD) analysis. The microstructure and pore structure of wood are characterized by scanning electron microscopy (SEM) and mercury intrusion analysis. The fire resistance properties of wood are tested and analyzed by a cone calorimeter (CONE). A 300 KN microcomputer-controlled electronic universal testing machine was used to test the compressive strength of the treated wood and study the impact of cyclic treatment on the mechanical properties of the wood. The research result is beneficial to the development of a wood flame-retardant treatment method.

## 2. Materials and Methods

### 2.1. Materials

Mongolian Scots pine (Pinus sylvestris Linn. var. mongolica Litv.) naturally occurs in the mountainous region of Daxing’anling, China. It was purchased from a local distributor in Huainan, China. The wood was cut into pieces and cuboids with sizes of 100 mm × 100 mm × 3 mm, 20 mm × 20 mm × 30 mm, and 20 mm × 20 mm × 300 mm, respectively. The calcium chloride (CaCl_2_), disodium hydrogen phosphate (Na_2_HPO_4_, DSP), and sodium hydroxide (NaOH) were analytically pure purchased from Tianjin Hengxing Chemical Reagent Manufacturing Co., Ltd. (Tianjin, China). The CaCl_2_ solution with a concentration of 0.5 mol/L was used to provide calcium ions. The DSP solution with a concentration of 0.5 mol/L was used to provide phosphate ions. Those two solutions were prepared and separately stored in plastic buckets. A small amount of NaOH (0.01 mol/L) was added to the DSP solution to form a high pH solution, which was conducive to the formation of phosphate precipitation.

### 2.2. Treating Method

The wood was dried in a drying chamber (DHG-9140B, Shanghai Peiyin Experimental Instrument Co., Ltd., Shanghai, China) at a temperature of 60 °C to constant weight. The dried wood was first immersed in a CaCl_2_ solution for 3 h to absorb calcium ions in the pores and then immersed in the DSP solution for another 3 h to allow for phosphate ions to reach the pores. The DSP can react with calcium ions to produce phosphate precipitate in the pores. The treated wood was removed from the solution and wiped with the surface with a towel. This processing was one cycle treatment as shown in Figure 1. For each cycle, the wood was immersed in a CaCl_2_ solution for 3 h and immersed in a DSP solution for another 3 h. After ten cycles of treatment, the treated wood was immersed in fresh water to remove the ion residue and unbounded precipitates. After this, the specimens were dried in a drying chamber with a temperature of 60 °C to constant weight. The dry-treated specimens were used for characterization and testing.

### 2.3. Characterization of Treated Wood

#### 2.3.1. Water Absorption and Bulk Density

Before being immersed in solution, the wood samples were weighed on a precision electronic scale (Yao Xin Dianzi Technology Co., Ltd. Shanghai, China) to measure the dry mass. After each cycle treatment, the sample was dried on the surface with a wet towel to remove the free water from the surface. Then, the surface dried sample was weighed immediately, which was recorded as M_s_(g). The sample was dried in a drying box at a temperature of 60 °C until the mass was constant. The dry mass M_d_(g) was measured after the sample was cooled down to room temperature. 

The water absorption (W_a_) was calculated according to the following equation: (1)$$\;\;W_{a}=\frac{M_{d}-M_{s}}{M_{d}}\;\;\;\;\;\;$$

The bulk density(ρ) was calculated according to the following equation:(2)$$\;\rho =\frac{M_{d}}{L\;\times \;W\;\times \;H}\;$$

Here: L (mm), W (mm), and H (mm) are the length, width, and height of the wood sample.

This part of the experiment processes 20 wood blocks at a time. The final result removes two maximum values and two minimum values and takes the average value as the final result.

#### 2.3.2. SEM, XRD, and MIP Test

A scanning electron microscope (SEM, Zeiss Gemini 500, Carl Zeiss, Jena, Germany) was used to observe the microstructure of wood. Each sample was coated with a thin layer of gold to avoid electron charging. A Smart lab SE type X-ray diffractor (XRD, Columbus, OH, USA) was used to detect the minerals of the precipitates deposited on the surface of treated wood. The detection range was from 5° to 70° with a scan range of 5°/min. The pore size distribution of wood can be probed by a MicroActive Autopore V9600 (Micromeritics Instruments Corporation, Norcross, GA, USA). The untreated wood, 5 treatments, and 10 treatments were used in the MIP test.

#### 2.3.3. Cone Calorimeter Test

A Cone Calorimeter test was carried out to characterize the combustion properties of the wood. The combustion behaviors of the samples were assessed by a cone calorimeter (VOUCH, 6810, Suzhou, China) according to the standard ISO5660-1 [30]. Squared specimens (100 mm × 100 mm × 3 mm) were horizontally exposed to a heat flux of 35 kW/m^2^.

#### 2.3.4. Mechanical Testing Test

The compressive strength test and the bending strength of wood were based on GB/T 1927–2021 [31]. After 1 treatment, 5 treatments, and 10 treatments, the wood was tested with a 300 KN microcomputer-controlled electronic universal testing machine (Sansi Zongheng Machinery Manufacturing Co., Ltd., Shanghai, China) for compressive strength and bending resistance. Among them, the number of woods used in each group of compressive strength tests was 3, and the dimensions were 20 mm × 20 mm × 30 mm. The compressive strength value obtained in each sample test was averaged as the final compressive strength. For a group of 3 pieces of wood with dimensions of 20 mm × 20 mm × 300 mm, the bending strength obtained after the bending test is averaged as the final bending strength.

## 3. Results and Discussions

### 3.1. Water Absorption Reduction and Bulk Density Increment

Figure 2 shows the water absorption and bulk density of treated wood for every cycle. The water absorption of untreated wood is 42.83%. When the wood was treated, water absorption was decreased by this treatment method. More reduction in water absorption can be achieved by more cycle treatment. As can be seen in Figure 2, the water absorption is 38.06% and 36.95% after 5 and 10 cycles of treatment, which are 11.13% and 13.75% lower than the untreated wood, respectively. The water absorption reduces significantly during cycles one to five. However, this reduction is gradually stable from 6 to 10 cycles, and the water absorption is stable after eight cycles of treatment. This indicates that there exists a maximum reduction in water absorption by this treatment method. 

The bulk density of wood is increased by the proposed treating method, indicating that the treated wood is denser than untreated wood. This is the main reason why water absorption is reduced by the proposed method. For the first five cycles, the bulk density is significantly increased and a 5.2% increase in bulk density can be reached, suggesting that the pores of the wood are filled. The trend of bulk density increment is slow after five cycles of treatment. After eight cycles of treatment, the bulk density is almost stable, indicating that the maximum bulk density is reached.

### 3.2. XRD Analysis

The XRD pattern of precipitation produced during the treatment is shown in Figure 3. The precipitation mainly contains hydroxyapatite (HAP), calcium hydrogen phosphate (CaHPO_4_·2H_2_O), calcium phosphate (Ca_3_(PO_4_)_2_), and sodium chloride (NaCl), which was in good agreement with previous studies [32,33]. The presence of NaCl is due to the residue of Na^+^ and Cl^−^ in treating solutions. The CaHPO_4_·2H_2_O was formed by HPO_4_^−^ binding with Ca^2+^. Calcium hydrogen phosphate is not stable under the alkaline condition, which can react with Ca^2+^ and OH^−^ to form calcium phosphate. The calcium phosphate can be transformed into HAP [34,35], as shown in Figure 3. The chemical reactions are as follows:(3)$${2\mathrm{HPO}}_{4}^{2^{-}}+{3\mathrm{Ca}}^{2+}+{2\mathrm{OH}}^{-}\to {{\mathrm{Ca}}_{3}({\mathrm{PO}}_{4})}_{2}↓+{2H}_{2}O$$
(4)$${{3\mathrm{Ca}}_{3}({\mathrm{PO}}_{4})}_{2}+{\mathrm{Ca}}^{2+}+{2\mathrm{OH}}^{-}\to {{{\mathrm{Ca}}_{10}({\mathrm{PO}}_{4})}_{6}(\mathrm{OH})}_{2}↓$$

### 3.3. SEM Analysis

In order to characterize the microscopic morphology of wood in the process of cycle treatments, the wood with different cycle treatments was tested by SEM. Figure 4 compares the microstructure of untreated and treated wood. In this figure, the untreated wood has a rough surface with some blowholes. Some precipitate is deposited on the surface of the wood after one cycle of treatment, which fills some pores. With the increase of cycle treatment, more precipitate is produced in the pores. After five cycles of treatment, the wood surface is covered with precipitate, as shown in Figure 4c. When 10 cycles of treatment are reached, more precipitations, which were the HAP, CaHPO_4_·2H_2_O, and Ca_3_(PO4)_2_, are deposited on the surface of the wood, as shown in Figure 4d. This is consistent with the microscopic morphology of phosphate prepared by Zeng et al. [36]. Figure 5b–d indicates that those precipitations are gradually transformed into globular particles, suggesting that the phosphate precipitation gradually transforms into a stable state. The CaHPO_4_·2H_2_O and Ca_3_(PO4)_2_ are gradually transformed into HAP [34,37], which is conducive to improving the fire resistance of wood [38].

### 3.4. Pore Size Distribution Analysis

Figure 5 compares the pore size distribution of untreated and treated wood. It can be seen from Figure 5a that the pore diameter of untreated wood is mainly distributed from 226 nm to 30,277 nm. After five cycles of treatment, the pore diameter of wood is mainly distributed in the pore size of 121 nm to 4928 nm, suggesting that large numbers of pores and blowholes are filled with precipitate as confirmed by SEM analysis. The pore diameter of 10 cycles of treated wood is mainly distributed in the pore sizes of 50 nm to 2693 nm, which attributes to more precipitate deposition. As shown in Figure 5b, the pore diameter of untreated wood is concentrated between 5000 nm (5 μm) and 40,000 nm (40 μm), which is caused by pores and blowholes in wood. Those pores and blowholes are converted to the small aperture as indicated by Figure 5b. The pore diameter of five cycles of treated wood is concentrated between 200 nm (0.2 μm) and 5000 nm (5 μm), and 10 cycles of treated wood is concentrated between 100 nm (0.1 μm) and 3000 nm (3 μm). The SEM analysis, XRD analysis, and MIP test show that phosphate precipitations are deposited on the surface of the wood and in the connecting pores on the surface, resulting in the reduction of wood porosity. Moreover, porosity decreases with the increase in cycle times. The cumulative pore volume of 10 cycles of treated wood is decreased by 54.3% compared to the untreated one, indicating the good potential of the proposed treatment method. 

### 3.5. Mechanical Testing Analysis

#### 3.5.1. Compressive Strength Analysis

The compressive strength of different cycles of treated wood is shown in Figure 6. It can be seen that the compressive strength of untreated wood can reach 49.6 MPa, which is consistent with previous research [39]. The compressive strength of the treated wood has increased to a certain extent, of which the strength of the recycled wood is basically the same as that of the untreated wood. This is mainly due to the small number of impregnations and the less phosphate deposited on the surface of the wood, which has also been confirmed in SEM analysis. The strength of the wood after five cycles increased by 7.38% compared with that of the untreated wood. This is due to the more phosphate deposited on the wood, which leads to the more encrypted structure of the wood pore. The compressive strength of wood treated in 10 cycles has increased by 14.7% compared to the untreated wood. It can be seen that with the increase in the number of cycle treatments, the impregnation efficiency has increased significantly. It has also been shown in detail in the SEM and MIP analysis.

#### 3.5.2. Bending Strength Analysis

The bending strengths of the wood at different cycles treated are shown in Figure 7. The bending strength of untreated wood is 88.2 MPa. It is basically consistent with the wood performance research parameters made by Atar [40]. The results show that the bending strength of the wood after one cycle treated is only 5.25% higher than that of the untreated wood. After five cycles of treatment, the bending strength of the wood increased by 11.2% compared to the untreated wood. It can be seen that multiple cycles can enhance the bending strength of the wood, which is mainly caused by the phosphate deposited on the wood, improving the wood pore structure, and making the wood more encrypted. This has been confirmed in the SEM and MIP tests. The bending strength of the treated wood after 10 cycles has been improved by nearly one step, which can reach 23.8%. This is mainly due to the increase of phosphate deposited on the wood and the improvement of the bending resistance of the wood. 

### 3.6. Cone Calorimeter Analysis

#### 3.6.1. Heat Release Rate and Total Heat Release

Heat release rate (HRR) and total heat release (THR) are usually used to measure the refractory performance of materials [41,42]. The higher HRR and THR indicate a high, intense, pyrolysis of the material, which is harmful to the burning wood and the environment [43,44]. Figure 8 shows the heat release rate (HRR) and total heat release (TRR) curves of different cycle-impregnated wood. It can be seen that the HRR curves of untreated wood have two heat release peaks. When the wood sample is ignited, it starts to burn and release heat, which is the reason for the presence of the first heat release peak. A carbon layer is formed on the surface of the wood sample. The carbon layer acts as a physical barrier, which can prevent heat and oxygen from penetrating into the sample [45,46,47]. However, due to the high porosity of the untreated wood sample, the carbon layer has no obvious barrier effect on the air. This leads to rapid internal combustion and a second heat release peak. As shown in Figure 8, the wood did not tend to stabilize after the first heat release peak and quickly rose to the second heat release peak. Those two heat release peaks also appeared in the five cycles of treated wood. However, these two peaks are smaller than the untreated sample. Up to 47.3% and 63.3% reductions in the first peak and second peak can be achieved by the five cycles of treatment. Meanwhile, the heat release peak can be significantly delayed after five cycles of treatment, as shown in Figure 8, leading to better performance of flame retardancy. This is attributed to two reasons: (one) the sample of five cycles-treated wood has a dense microstructure, which can effectively prevent heat and oxygen from entering the sample with the carbon layer; (two) phosphate precipitates deposited on the wood are good flame-retardant materials. These precipitates not only have good thermal stability, they also absorb a lot of heat during the separation of bound water. It has also been proven to be a flame retardant in previous studies [48]. The total heat release of the five-cycles-treated wood is decreased by 7.8% compared to the untreated one. The HRR curve of wood with 10 cycles of treatment has only one peak with 127.8 kW/m^2^. Moreover, this sample did not release heat in the first 41 s, indicating that the wood did not start to burn during this period. More importantly, the heat release rate is slow after 71 s, suggesting that the treated wood has excellent flame retardancy. This is attributed to two reasons: (one) more phosphate deposits make the wood surface more difficult to ignite; (two) the dense carbon layer can block the internal combustion, leading to the significant reduction of heat release peak. Since the excellent flame retardancy, the total heat release of the 10-cycles-treated wood was reduced by 48.7% compared to the untreated wood.

#### 3.6.2. Smoke Production Rate and Total Smoke Production

The smoke production rate (SPR) and the total smoke production (TSP) are the main parameters to evaluate the fire safety performance of the materials [49]. The results of SPR and TSP are shown in Figure 9. Clearly, all samples have two smoke production emission peaks. For the untreated wood, the maximum SPR of the first peak is reached 0.0311 m^2^/s at 31 s, which is higher than other treated samples. After the initial ignition and combustion test, the wood is immediately pyrolyzed, which is mainly carried out on the surface of the wood. The wood is gradually decomposed into small molecular combustible gases, such as tar and coke. The pyrolysis process absorbs part of the heat, which results in less heat energy on the wood surface and more noncombustible volatiles, and incompletely oxidized organics [50,51]. This is the main reason for the rapid increase in the smoke production rate from untreated wood in this short period of time. After the cycle treatment, phosphate deposits are deposited on the surface of the wood, which inhibits the decomposition of the wood into small combustible gases to a certain extent. These phosphate precipitates have a protective effect on the wood, resulting in lower smoke emissions. The test results show that the first peak of the five-cycles-treated wood is reduced by 72.3% compared to the untreated wood. As the combustion continues, the surface temperature of the wood continues to increase, and the combustion begins to proceed inward. Since combustion is a very violent physicochemical reaction, the ashes burned on the surface of the wood can volatilize into the air. Some of the unoxidized small particles produced by insufficient combustion will be released with heat dissipation [52]. This is the main reason for the second peak of SPR. For the five-cycles-treated wood, the carbon layer formed is denser because the wood is catalyzed into a carbon better, leading to the adsorption of more smoke and volatile small particles. With the increase of cycles treatment, some phosphate can be deposited inside the wood, which hinders the combustion inside the wood. After 10 cycles of treatment, the second smoke emission peak of the combustion process is significantly reduced compared to the untreated wood lower than that of the control group. However, the second smoke emission peak of the wood treated with 10 cycles is slightly higher than that of the samples treated with five cycles. This is due to the decomposition of phosphates inside the wood, resulting in the formation of small volatile particles [53]. Since combustion is hindered by the presence of phosphates, the TSP of the treated wood is significantly reduced, as shown in Figure 8. The results showed that the TSP of wood after 5 and 10 cycles of treatment is 42.9% and 54.2% lower than that of the untreated wood, respectively. Compared with previous relevant research [54], the results of SPR and TSP indicate that the fire safety of the wood can be significantly improved by the proposed treatment method.

## 4. Conclusions

This study proposed a new method to treat Mongolian Scots pine wood. This method precipitates phosphate precipitations in the pores of the wood, which is conducive to improving the fire resistance of the wood. The experimental study shows that:

(1)Water absorption of wood is significantly reduced and mechanical properties are improved by the cycles treatment, which attributes to the precipitated phosphate precipitations filling in the pores of the wood. For this reason, up to 54.3% of redaction on the pore volume, 14.7% of enhancement on the compressive strength, and 23.8% of improvement in the bending strength of wood can be achieved by the 10 cycles treatment;(2)The phosphate precipitations can act as a flame-retardant layer to improve the refractory of the wood. This flame-retardant layer delays the infiltration of heat and oxygen during combustion. It not only reduces the total heat release, it also delays the time and intensity of the heat release peak;(3)The deposited phosphates inhibit the decomposition of wood into small combustible particles. This protection reduces the smoke release rate and total release from the wood. The TPR is greatly reduced by the proposed method. A more than 54% reduction in TSP demonstrates the improvement in the fire safety of cycles-treated wood.

In this study, the proposed method not only enhances the mechanical properties of wood, it also improves the fire resistance of wood, which is of great significance to the fire protection of wood products. Phosphate precipitation is nontoxic, harmless, and eco-friendly. In theory, this method has good applicability and is also effective for other wood and oxidized and corroded wood. It should be pointed out that this study is just a proof of concept of the proposed method. More research work will be carried out in the future.

## Figures and Tables

**Figure 1 materials-16-02711-f001:**
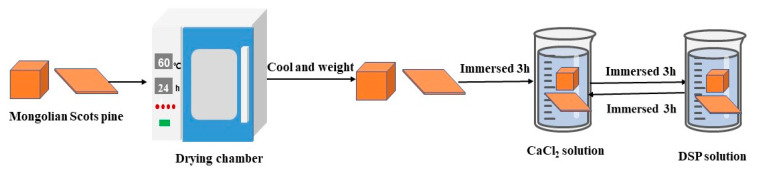
Schematic diagram of Mongolian Scots pine treatment process.

**Figure 2 materials-16-02711-f002:**
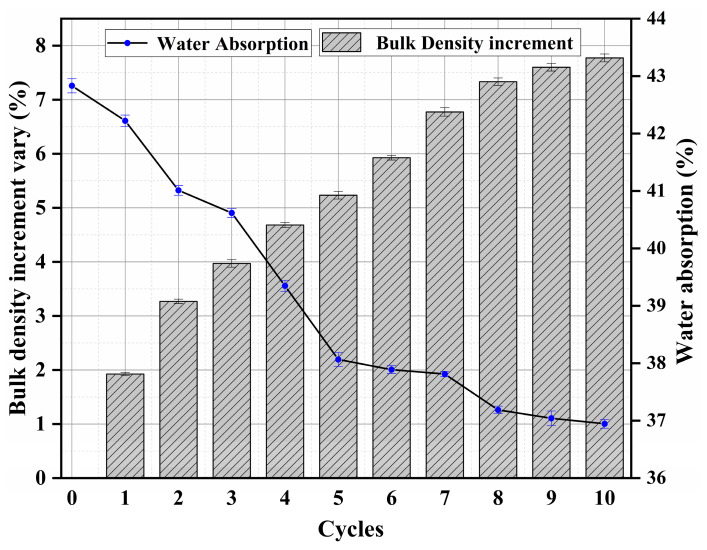
Water absorption and bulk density increment vary with the number of cycles of treatment.

**Figure 3 materials-16-02711-f003:**
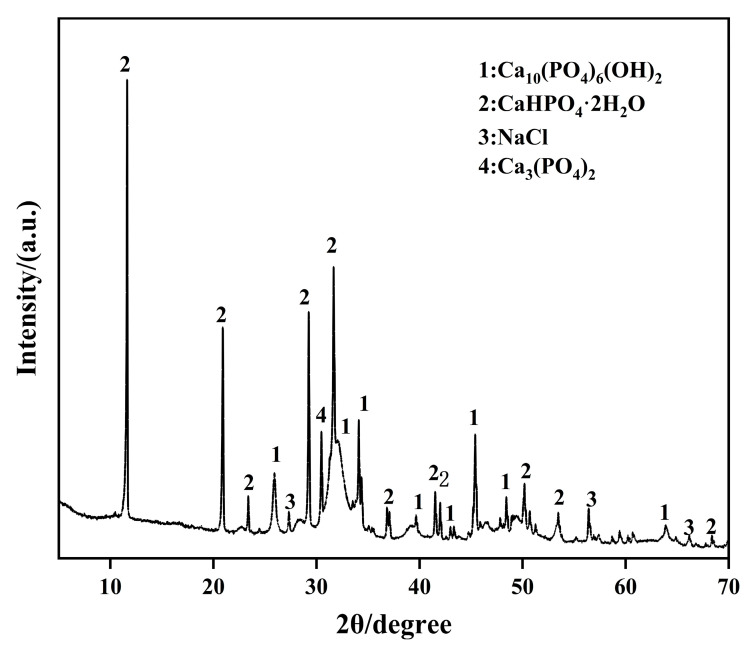
XRD patterns of the precipitations on the wood surface.

**Figure 4 materials-16-02711-f004:**
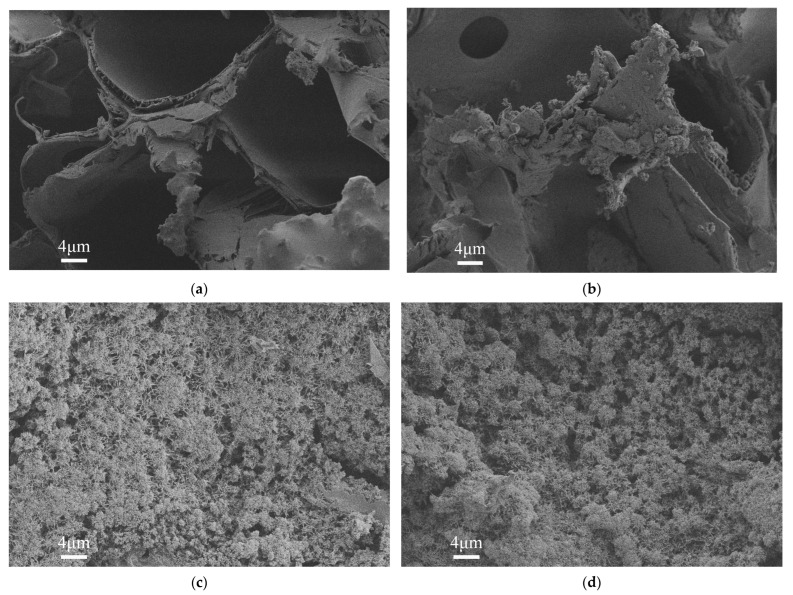
SEM images of the wood samples at 2000× magnification: (**a**) Untreated; (**b**) 1 cycle treatment; (**c**) 5 cycles treatment; (**d**) 10 cycles treatment.

**Figure 5 materials-16-02711-f005:**
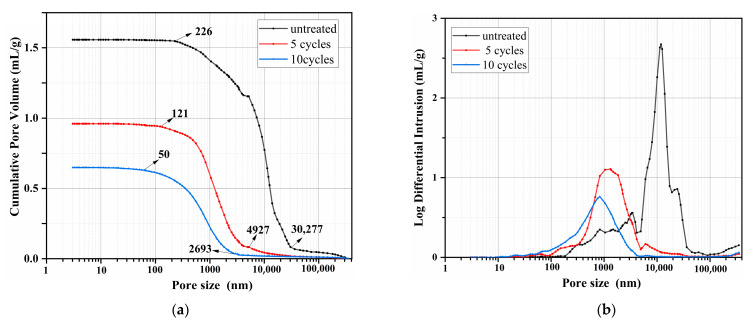
Pore size distribution of untreated and treated wood samples: (**a**) cumulative pore volume; (**b**) pore size distribution.

**Figure 6 materials-16-02711-f006:**
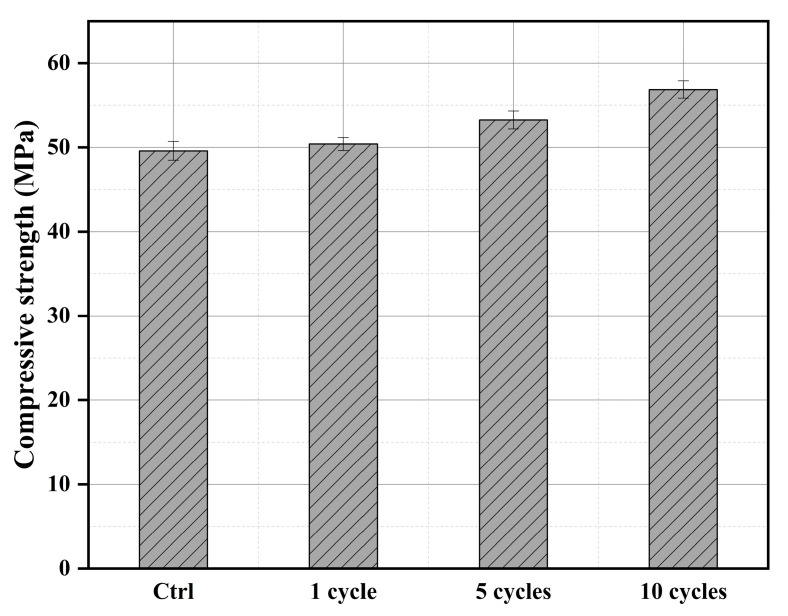
Compressive strengths of the Mongolian Scots pine at different cycles treated.

**Figure 7 materials-16-02711-f007:**
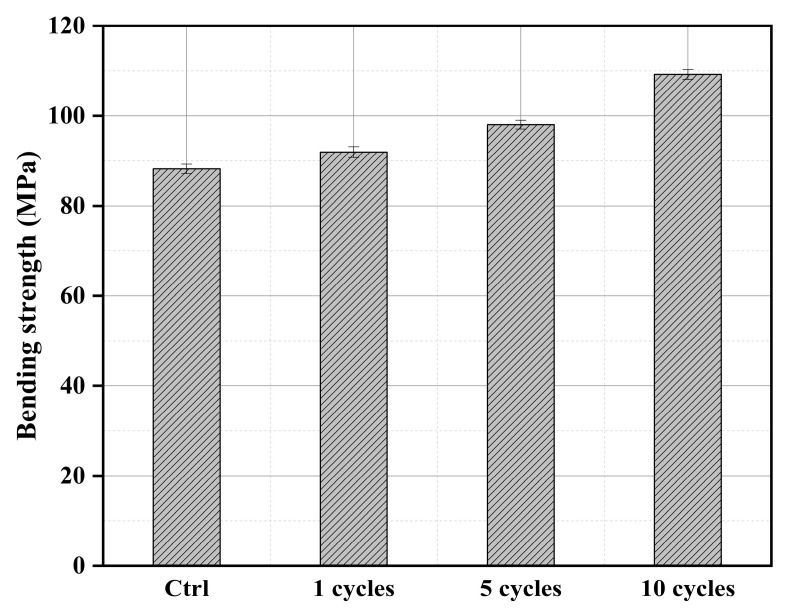
Bending strengths of the Mongolian Scots pine at different cycles treated.

**Figure 8 materials-16-02711-f008:**
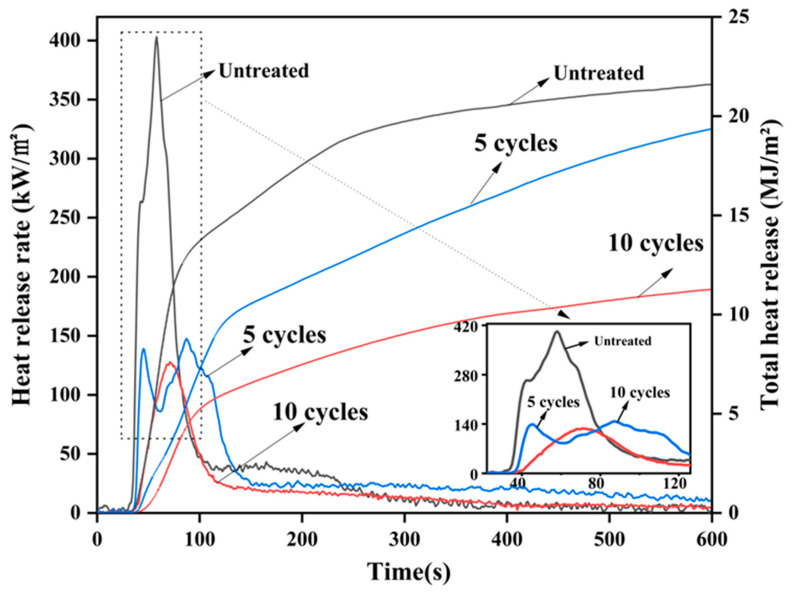
The heat release rate and total heat release of untreated and treated wood.

**Figure 9 materials-16-02711-f009:**
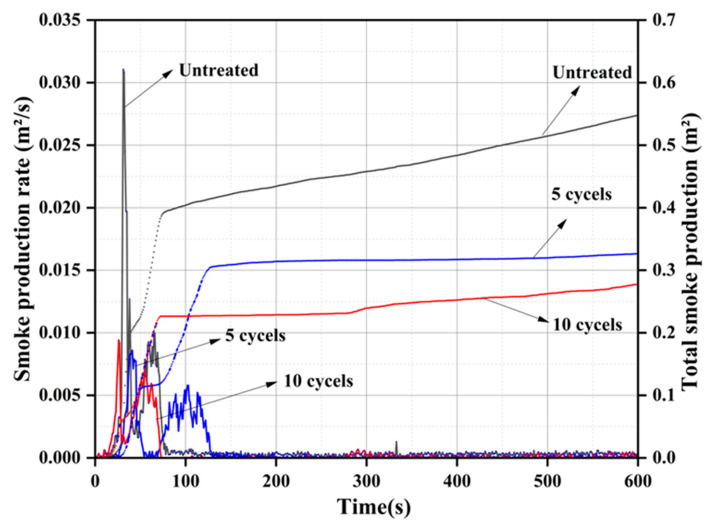
Smoke production rate and total smoke production of untreated and treated wood.

## Data Availability

The study did not report any data.

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
