# Peer review of "Surface Treatment of Mongolian Scots Pine Using Phosphate Precipitation for Better Performance of Compressive Strength and Fire Resistance"

_materials, 2023, doi:10.3390/ma16072711_

Round 1

Reviewer 1 Report

The authors present a work on the treatment of wood with ceramic substances to improve their properties in specific applications. The document is well organized and structured in its sections and content. The references are considered current, but they are limited and could be increased in quantity.

In the materials section, it is important to include the origin of the materials used, as this allows the reproducibility of the experiments.

In the first paragraph of section 2.2, what do the authors mean by the phrase: "at constant weight"?

It is considered important to include the references of identification of substances by XRD.

The description of the SEM and XRD analysis equipment should be included.

How was the porosity distribution obtained in the samples?

It is important to describe the analysis methodology to measure HRR, THR, TRR, SPR and TSP; or, cite the corresponding rule.

Author Response

Dear reviewers, thank you very much for your suggestions and comments. Please see the attachment

Reviewer 2 Report

I reviewed article entitled “Surface Treatment of Wood Using Fire-resistance Mineral Precipitation” by Yan Ge et al. This is a straight forward experimental article which falls within the scope of your journal. I have following questions and comments for the authors, once the manuscript is revised based on them I would expect it will be ready for publication.

1.       In Introduction, I would like to see more solid findings on the effect of surface treatment on the mechanical properties of the wood.

2.       In the 2.3 Characterization of treated wood, how many replications were conducted on each test? Standard deviation of the average results should also show in Figures 2 and 3.  

3.       To make the manuscript more concise, Figures 2 and 3 could be combined.

4.       Line 156, “For the first 5 cycles, the bulk density is significantly increased 154 and 5.2% increase in bulk density can be reached, suggesting that the pores of wood are filled.” Show the evidence in SEM to support your suggestion.

5.       Recheck Line 181 “as shown in Figure 5(d). Figure 5(c) and Figure 5(d)”

6.       Line 214 - Specify the magnification in Figure 5 caption.

7.       In Results and discussions, lack of comparison and support with the literature reviews.

8.       Mechanical testing is recommended to fulfill the completion of the manuscript.

Author Response

Dear reviewers, thank you very much for your suggestions and comments. Please see the attachment.

Reviewer 3 Report

The manuscript deals with investigation on surface treatment of pine wood by phosphate precipitation, aimed at improving its fire resistance. Overall, the manuscript is well-written, structured and informative, but has to be further revised before acceptance for publication. Please, see below my comments on your work:

The title is too general and does not provide any specific information about the specific aims of your research, please revise it.

The abstract (lines 9 to 27) and the keywords (line 28) correspond to the aims and objectives of the manuscript.

The abstract is concrete and informative, and contains the main findings of the article.

In the keywords, I’d recommend to add also ‘fire resistance’.

Lines 72-76: the statements provided are generally true, and considering the increased environmental awareness and the growing demand for more bio-based products, including fire retardants, I think it would be beneficial to add a short paragraph on this issue. Please check the following relevant references:

https://doi.org/10.1016/j.polymdegradstab.2022.110153

https://doi.org/10.1016/j.mser.2017.04.001

https://doi.org/10.3390/polym14030362

https://doi.org/10.1016/j.ijbiomac.2022.01.007

https://doi.org/10.1021/acssuschemeng.6b00112

Overall, the Introduction part is well written and informative, and provides relevant information on the topic of the research, based on previously published studies. The addition of more references will increase the scientific soundness of the paper.

Line 90: ‘pine wood’ – please be more specific, and provide the tree species used, including its botanical name.

Please be more specific also about the suppliers of both wood material and chemicals used.

Line 111, Figure 1: there is ‘camphor wood’ in the figure, please explain/revise. In addition, it is written ‘dring box’, may be ‘drying chamber’ as mentioned above. What is the duration of 1 cycle treatment?

Line 114: please provide relevant information about the electronic scale used (company producer, city, country).

Line 124: please add the missing dimension unit.

Line 126: it should be ‘scanning electron microscope’, please revise. In addition, please add relevant information about the equipment used (company, city, country).

Line 132: Please add relevant information about the cone calorimeter used.

Overall, the Materials and Methods section is well prepared, but should be further elaborated based on the above comments.

The results of the study are detailed and informative. However, a more detailed discussion with previous studies in the field is required.

The Conclusion part reflects the main findings of the research. Here I’d recommend to add how the results of your research work can be integrated into industrial practice, as well as the potential of future studies in this field.

The references cited are appropriate and correspond to the topic of the manuscript. However, their number (only 17) is not sufficient. Please add more relevant references, especially in the Introduction and Results and Discussion section. 

Author Response

(The authors gave the same response as above.)

Round 2

Reviewer 3 Report

Dear authors, 

Thank you for addressing all my previous comments/remarks. I have one small additional comment: the most common commercial name of Pinus sylvestris is Scots pine (UK). Why did you use "camphor pine"?
